# Water-Soluble Ions in Atmospheric Aerosol Measured in a Semi-Arid and Chemical-Industrialized City, Northwest China

**Huimin Jiang [1], Zhongqin Li [1,2,3,*], Feiteng Wang [2], Xi Zhou [4], Fanglong Wang [2], Shan Ma [4] and Xin Zhang [2]**

[1] College of Geography and Environmental Science, Northwest Normal University, Lanzhou 730000, China; jianghm0201@163.com
[2] State Key Laboratory of Cryospheric Sciences, Northwest Institute of Eco-Environment and Resources, Tianshan Glaciological Station, Chinese Academy of Sciences, Lanzhou 730000, China; wangfeiteng@lzb.ac.cn (F.W.); wangfanglong@nieer.ac.cn (F.W.); zhangxin@nieer.ac.cn (X.Z.)
[3] College of Sciences, Shihezi University, Shihezi 832000, China
[4] Key Laboratory of Western China's Environmental Systems (Ministry of Education), College of Earth and Environmental Sciences, Lanzhou University, Lanzhou 730000, China; zhouxi13_lzu@163.com (X.Z.); mashan706@126.com (S.M.)
[*] Correspondence: lizq@lzb.ac.cn

**Abstract:** We investigated water-soluble ions (WSIs) of aerosol samples collected from 2016 to 2017 in Lanzhou, a typical semi-arid and chemical-industrialized city in Northwest China. WSIs concentration was higher in the heating period ($35.68 \pm 19.17$ μg/m$^3$) and lower in the non-heating period ($12.45 \pm 4.21$ μg/m$^3$). $NO_3^-$, $SO_4^{2-}$, $NH_4^+$ and $Ca^{2+}$ were dominant WSIs. The concentration of $SO_4^{2-}$ has decreased in recent years, while the $NO_3^-$ level was increasing. WSIs concentration was affected by meteorological factors. The sulfur oxidation and nitrogen oxidation ratios (SOR and NOR) exceeded 0.1, inferring the vital contribution of secondary transformation. Meanwhile higher $O_3$ concentration and temperature promoted the homogeneous reaction of $SO_2$. Lower temperature and high relative humidity (RH) were more suitable for heterogeneous reactions of $NO_2$. Three-phase cluster analysis illustrated that the anthropogenic source ions and natural source ions were dominant WSIs during the heating and non-heating periods, respectively. The backward trajectory analysis and the potential source contribution function model indicated that Lanzhou was strongly influenced by the Hexi Corridor, northeastern Qinghai–Tibetan Plateau, northern Qinghai province, Inner Mongolia Plateau and its surrounding cities. This research will improve our understanding of the air quality and pollutant sources in the industrial environment.

**Keywords:** total suspended particulates (TSP); water-soluble ions (WSIs); formation mechanism; source identification; Lanzhou

## 1. Introduction

Due to rapid urbanization processes and industrialization development, aerosols have drawn much attention from the government, the public, and scientists. Aerosols, particularly anthropogenic aerosols, have adverse impacts on atmosphere visibility, climate change, and human health [1–3]. Water-soluble ions (WSIs) are essential components of aerosols, which include sulfate ($SO_4^{2-}$), nitrate ($NO_3^-$), ammonium ($NH_4^+$) and alkali cations. They can contribute to 60–70% of the total suspended particles (TSP) [4]. They not only play a significant role in changing the earth's radiation balance but also promote the formation of acid rain [5]. Moreover, $SO_4^{2-}$, $NO_3^-$ and $NH_4^+$ are secondary ions of WSIs, which would affect the acidity of TSP and accelerate hazardous particulate materials [6]. In recent years, a series of studies on the chemical and physical properties of aerosols have been conducted in many countries [7–10]. Moreover, China has begun to study them since the early 1990s [11–13]. For the past few years, China's air quality has been much lower than that of other countries, which has remained in the spotlight in

many cities in China, like Xi'an, Shanghai, Dalian, Beijing and other developed cities and studies have been conducted to characterize WSIs and their contribution to TSP [14–17].

Lanzhou, located in the arid and semi-arid areas of the northwest, is a crucial economic hub of the Silk Road and the largest chemical base in northwest China. Due to the particular natural environment and dust storms over the past few decades, Lanzhou is facing serious air quality problems [18]. Nevertheless, Lanzhou has promulgated a series of strict policies and management measures to curb air pollution such as emission reduction, dust suppression, vehicle control. However, the studies of WSIs in valley-industrialized cities of China are relatively deficient, such as in Lanzhou where the TSP pollution is severe. A lot of previous studies have investigated air pollution, mainly focused on the concentration and chemical components of TSP in seasons [19–24]. The source apportionment of the typical pollutants that exceed the standard over Lanzhou during the heating and non-heating periods is still unclear.

The aims of this study are to (1) characterize the variation of WSIs during the heating and non-heating periods in Lanzhou. (2) Investigate the influence of natural meteorological factors on WSIs, identify and quantify the secondary transformation of $SO_4^{2-}$ and $NO_3^-$. (3) Compare the long-term WSIs and track the sources of atmospheric pollutants in Lanzhou. The results would help us not only to better understand the chemical composition and potential sources of TSP in the heating and non-heating periods in Lanzhou but also to provide useful information for establishing control strategies of aerosol pollution.

## 2. Experimental Site and Methodology

### 2.1. Sampling Site Description

Lanzhou is surrounded by mountains and situated in a narrow river valley and that leads to its special meteorological conditions and inversion layer. The mean annual wind speed (WS) is only 0.8 m s$^{-1}$ and the inversion weather accounts for about 80% of the days of the year [25]. Lanzhou is divided into four districts: Xigu, Anning, Qilihe, and Chengguan. Since the Chengguan district in Lanzhou is the center of government, residence, and commerce, several mixed pollution sources might be found. The sampling site was located in the Chengguan District of Lanzhou (Figure 1). The instrument was mounted on the roof of Scientific Research building NO.1 at the Northwest Institute of Eco-Environment and Resources (NIEER) (103.86° E, 36.05° N), and was about 20 m above the ground surface. At the same time, there are no visible high-rise buildings within 2 km around the sampling site. The sampling site is sheltered from dust and represents the typical atmospheric urban environment in Lanzhou.

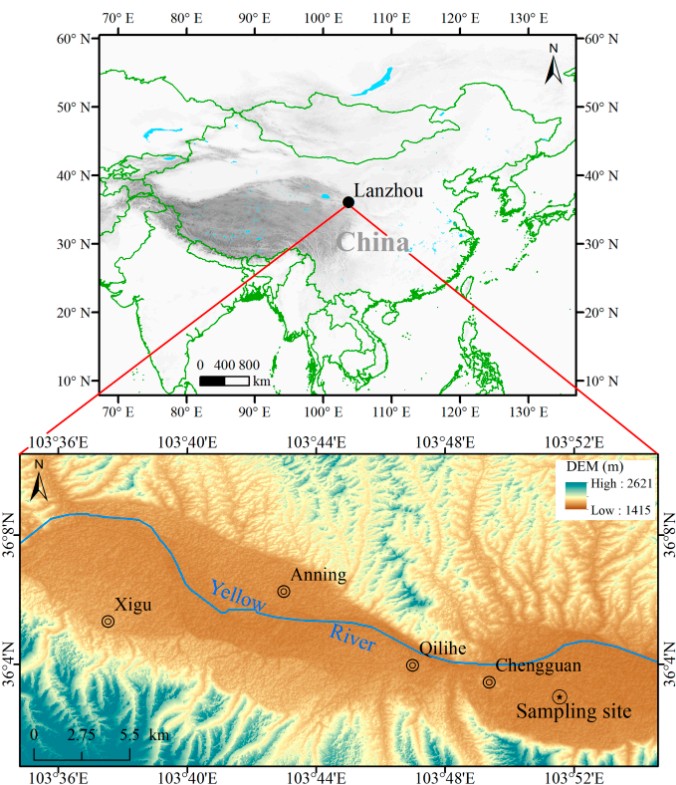

**Figure 1.** The geographic location of Lanzhou.

### 2.2. Chemical Collection and Analysis

In total, 42 TSP filter samples were collected on Teflon® Zefluor™ filters (47 mm in diameter with 2 µm pore size) using a 12-V diaphragm pump powered by solar cells. Aerosol samples were collected from December 2016 to October 2017. Among them, 18 samples were collected in the heating periods (December 2016–February 2017) and 24 samples were collected in the non-heating periods (April–October 2017). Each sampling episode was conducted continuously from 21:00 and lasted for 22 h. The air volume passing through the filter was measured by an in-line meter and then converted into standard conditions according to the local environmental pressure and temperature. Based on the average flow rate of 1.27 $m^3 h^{-1}$ on the filter, the collection efficiency of particles was greater than 97% for aerosol particles with a diameter larger than 0.035 µm [26]. The five field blank filters were also collected without ambient air passing through the instrument during sampling periods. Both samples and field bland filters were stored at 4 °C during field sampling transportation, before the laboratory analysis in the State Key Laboratory of Cryospheric Science (SKLCS) of NIEER, Chinese Academy of Sciences (Lanzhou). All the filters were weighed at least three times before and after sampling with an accuracy of 0.1 mg in a class 100 clean room. In order to efficiently extract the water-soluble species from the filters, the sample and blank filters were first wetted with 200 µL ultra-pure methanol and then extracted with 25 mL of deionized water for about 30 min by an ultrasonic water bath device [27–29].

Major ions (including: $SO_4^{2-}$, $NO_3^-$, $Na^+$, $K^+$, $NH_4^+$, $Ca^{2+}$, $Mg^{2+}$ and $Cl^-$) were analyzed using an ion chromatograph (Dionex-320, Thermo Fisher, US) which consists of a guard column (CG12A for cation and AG11-HC for anion), an analytical column (CS12A for cation and AS11-HC for anion), a suppressor (CAES for cation and ASRS-4 mm for anion), a curb (28 mA for cation and 59 mA for anion) and an eluent (15 $mmol·L^{-1}$ MSA for cation and 15 $mmol·L^{-1}$ NaOH for anion) [28]. The concentration of eight major WSIs was calculated by subtracting the blank filters $Cl^-$ (3.66 ng/g), $NO_3^-$ (17.31 ng/g), $SO_4^{2-}$ (2.60 ng/g), $Na^+$ (2.61 ng/g), $NH_4^+$ (3.35 ng/g), $K^+$ (0.04 ng/g), $Mg^{2+}$ (2.44 ng/g), and $Ca^{2+}$

(6.41 ng/g), respectively. The detection limits of ions were within the range of 0.01 to 0.03 $\mu g/m^3$ and the relative standard deviation of each ion was <5%.

### 2.3. Data Resource

The meteorological parameters were acquired from an automatic weather station (AWS), which involved atmospheric temperature, pressure, relative humidity (RH), WS, and wind direction. The annual exhaust emissions, primary energy sources of industries and car ownership in recent years were from the Lanzhou Statistical Yearbooks (http://tjj.lanzhou.gov.cn/index.html accessed on 31 March 2021). The hourly mean concentration of Air Quality Index (AQI) and normative pollutants ($SO_2$, $NO_2$, CO, $O_3$, particulate matter $(PM)_{2.5}$, $PM_{10}$) from 1 December 2016 to 31 October 2017 were from the China National Environmental Monitoring Center (http://106.37.208.233:20035/ accessed on 31 March 2021).

### 2.4. Calculation of SOR and NOR

The Sulfur oxidation (SOR) and nitrogen oxidation ratio (NOR) were used to measure the secondary aerosol transformation efficiency of $SO_2$ to $SO_4^{2-}$ and $NO_2$ to $NO_3^-$, respectively, and they can be estimated using the following formulate [30]:

$$SOR = \frac{[SO_4^{2-}]}{[SO_4^{2-}] + [SO_2]} \tag{1}$$

$$NOR = \frac{[NO_3^-]}{[NO_3^-] + [NO_2]} \tag{2}$$

where $[SO_4^{2-}]$, $[SO_2]$, $[NO_3^-]$ and $[NO_2]$ are the molar concentrations of $SO_4^{2-}$, $SO_2$, $NO_3^-$ and $NO_2$, respectively.

### 2.5. HYSPLIT Model Analysis

To determine the potential long-distance transport pathways of air mass reaching Lanzhou, 72 h backward air-mass trajectories were calculated using the Hybrid Single-Particle Lagrangian Integrated Trajectory (HYSPLIT) model and the National Centre for Environmental Prediction (NCEP/NCAR) Global Data Assimilation System (GDAS) dataset (ftp://arlft.Arlh.Noaa.gov/pub/Archieve accessed on 31 March 2021). We obtained a total of 365 daily trajectories averaged from 6-h interval trajectories (including 00:00, 06:00, 12:00 and 18:00), which can be grouped into three clusters via the built-in clustering tool in the model. We also calculated the median trajectory for each cluster. More details of the cluster analysis are available in Draxler et al. [31].

The potential source contribution function (PSCF) and concentration weighted trajectory analysis (CWT) were used to identify the potential sources of pollution by air-mass trajectories [32]. The PSCF was used to calculate the probability that a source was located at latitude $i$ and longitude $j$ and the values were calculated using the following equation [33]:

$$p = \frac{m_{ij}}{n_{ij}} \tag{3}$$

where $m_{ij}$ is the total number of trajectories leading to the concentration of pollutants exceeding a criterion value, $n_{ij}$ is the number of trajectories that go through the cells. When the $n_{ij}$ is relatively low, the PSCF value has higher uncertainty, the $W_{ij}$ has been proposed consequently. If the $n_{ij}$ in a cell is less than three times the average number of endpoints in the grid, any weight function can be multiplied $W_{ij}$, to reduce the deviation of small $n_{ij}$ values. $W_{ij}$ was denoted as follows [34]:

$$W_{ij} = \begin{cases} 1 & 80 < n_{ij} \\ 0.70 & 20 < n_{ij} \leq 80 \\ 0.42 & 10 < n_{ij} \leq 20 \\ 0.05 & n_{ij} \leq 10 \end{cases} \tag{4}$$

However, the PSCF shows the proportion of polluted trajectories in the cells unilaterally. When the PSCF value is identical, the contribution to the receptor site cannot be distinguished. Concentration weighted trajectory analysis (CWT) is a method to identify potential sources and their relative importance, which overcome the limitation of PSCF [35]. In this method, the weighted concentration of each grid cell in a potential source area is calculated, which is acquired by averaging the concentration of the corresponding trajectory that crossed the grid cell:

$$C_{ij} = \frac{\sum_{l=1}^{M} C_l \cdot \tau_{ijl}}{\sum_{l=1}^{M} \tau_{ijl}} W_{ij} \tag{5}$$

where $C_{ij}$ is denoted the average weight concentration of the $ij$th cell, $l$ is the index of the trajectory, M is the total numbers of the grid $(i, j)$, $C_l$ is the mass concentration obtained from the receptor site $l$, $\tau_{ijl}$ is the time that the trajectory $l$ stays in the $ij$th cell. The higher the $C_{ij}$ values, the higher the association between air parcels traveling over the $ij$th cell and the concentration at the receptor.

## 3. Results and Discussion

### 3.1. Overall WSIs Concentration

Table 1 shows the concentration of WSIs during the heating and non-heating periods in Lanzhou. WSIs concentration ranged from 12.84 to 89.99 $\mu g/m^3$ during the heating period, with an average of 35.68 ± 19.17 $\mu g/m^3$. The rank of ion concentrations measured in the heating period was: $NO_3^-$ (12.20 ± 9.69 $\mu g/m^3$) > $SO_4^{2-}$ (6.83 ± 3.47 $\mu g/m^3$) > $Ca^{2+}$ (6.82 ± 2.93 $\mu g/m^3$) > $NH_4^+$ (5.84 ± 2.77 $\mu g/m^3$) > $Cl^-$ (2.11 ± 1.12 $\mu g/m^3$) > $Na^+$ (0.84 ± 0.34 $\mu g/m^3$) > $K^+$ (0.65 ± 0.35 $\mu g/m^3$) > $Mg^{2+}$ (0.39 ± 0.14 $\mu g/m^3$), which indicated that $NO_3^-$, $Ca^{2+}$, $SO_4^{2-}$ and $NH_4^+$ were the main WSIs in Lanzhou, accounting for 88.81% of WSIs. WSIs concentration ranged from 3.96 to 20.49 $\mu g/m^3$ during the non-heating period, with an average of 12.45 ± 4.21 $\mu g/m^3$. The rank of ion concentrations measured in the non-heating period was: $Ca^{2+}$ (4.37 ± 2.20 $\mu g/m^3$) > $SO_4^{2-}$ (3.27 ± 1.20 $\mu g/m^3$) > $NO_3^-$ (2.41 ± 1.48 $\mu g/m^3$) > $NH_4^+$ (1.04 ± 0.72 $\mu g/m^3$) > $Cl^-$ (0.51 ± 0.23 $\mu g/m^3$) > $Na^+$ (0.41 ± 0.50 $\mu g/m^3$) > $K^+$ (0.24 ± 0.10 $\mu g/m^3$) > $Mg^{2+}$ (0.21 ± 0.11 $\mu g/m^3$). The concentration of WSIs during the heating period was higher than in the non-heating period, and the main WSIs ($NO_3^-$, $Ca^{2+}$, $SO_4^{2-}$ and $NH_4^+$) in the non-heating period were the same as in the heating period, accounting for 88.99% of WSIs.

**Table 1.** The concentration of WSIs in Lanzhou ($\mu g/m^3$).

| Events | Project | $Cl^-$ | $NO_3^-$ | $SO_4^{2-}$ | $Na^+$ | $NH_4^+$ | $K^+$ | $Mg^{2+}$ | $Ca^{2+}$ | Total Ions |
|---|---|---|---|---|---|---|---|---|---|---|
| Heating period | Mean value | 2.11 | 12.20 | 6.83 | 0.84 | 5.84 | 0.65 | 0.39 | 6.82 | 35.68 |
| | Maximal value | 4.34 | 40.51 | 18.18 | 1.68 | 11.85 | 1.42 | 0.71 | 13.38 | 89.99 |
| | Minimum value | 0.52 | 2.07 | 3.12 | 0.27 | 0.64 | 0.19 | 0.13 | 2.48 | 12.84 |
| | Percentage/% | 5.93 | 34.18 | 19.15 | 2.35 | 16.37 | 1.81 | 1.10 | 19.11 | 100.00 |
| | STD | 1.12 | 9.69 | 3.47 | 0.34 | 2.77 | 0.35 | 0.14 | 2.93 | 19.17 |
| Non-heating period | Mean value | 0.51 | 2.41 | 3.27 | 0.41 | 1.04 | 0.24 | 0.21 | 4.37 | 12.45 |
| | Maximal value | 1.02 | 5.89 | 5.24 | 2.67 | 2.18 | 0.53 | 0.56 | 9.18 | 20.49 |
| | Minimum value | 0.19 | 0.35 | 1.03 | 0.03 | 0.18 | 0.09 | 0.05 | 0.80 | 3.96 |
| | Percentage/% | 4.07 | 19.33 | 26.22 | 3.31 | 8.36 | 1.94 | 1.69 | 35.07 | 100.00 |
| | STD | 0.23 | 1.48 | 1.20 | 0.50 | 0.72 | 0.10 | 0.11 | 2.20 | 4.21 |

Total ions: $[Na^+] + [NH_4^+] + [K^+] + [Mg^{2+}] + [Ca^{2+}] + [Cl^-] + [NO_3^-] + [SO_4^{2-}]$. Percentage: the ratio of corresponding ions to the total ion concentration.

$NO_3^-$ and $SO_4^{2-}$ are the two most important anions in the TSP samples in Lanzhou. $NO_3^-$ mainly represents mobile sources, such as vehicular exhausts. The high concentration of $NO_3^-$ had been observed in many cities, which indicated that vehicle emissions contribute a lot to $NO_x$ [11]. $SO_4^{2-}$ mainly represents stationary sources, such as fossil fuels, coal combustion, biomass combustion and transportation [36,37]. As shown in Table 1, the concentrations of $NO_3^-$ and $SO_4^{2-}$ during the heating-period ($12.20 \pm 9.69$ $\mu g/m^3$ and $6.83 \pm 3.47$ $\mu g/m^3$) were higher than those during the non-heating-period ($2.41 \pm 1.48$ $\mu g/m^3$ and $3.27 \pm 1.20$ $\mu g/m^3$), there might be a large number of primary pollutants (such as $SO_2$ and $NO_x$) produced by coal combustion during the heating period. In addition, there were relatively stable atmospheric conditions during the heating period and pollutants found it difficult to diffuse. During the heating period, the average concentration of $NO_3^-$ was about twice that of $SO_4^{2-}$, which may be due to the mobile sources playing a leading role in atmosphere, and the secondary transformation of $NO_2$ was also more effective. In contrast, $NO_3^-$ and $SO_4^{2-}$ dropped sharply during the non-heating period, and the concentration of $SO_4^{2-}$ exceeded that of $NO_3^-$, which proved that the contribution of $NO_3^-$ and $SO_4^{2-}$ to air pollution is alleviated. The increase in photochemical activity was one of the important reasons why $SO_4^{2-}$ was enhanced during the non-heating period, becoming the main secondary source.

$Ca^{2+}$ and $NH_4^+$ are the two most important cations. $Ca^{2+}$ is one of the best tracers for soil and construction dust with the highest concentration frequency observed in spring when a dust storm occurred [17]. $Ca^{2+}$ ($6.82 \pm 2.93$ $\mu g/m^3$) was the highest cation, followed by $NH_4^+$ ($5.84 \pm 2.77$ $\mu g/m^3$) during the heating period. $NH_4^+$ is more likely related to the decomposition of fertilizers and local sanitary waste [38]. Seinfeld has demonstrated that the formation of $NH_4^+$ only occurred under low temperature, $NH_4^+$ is transformed back to $NH_3$ at high temperature [39]. $Ca^{2+}$ ($4.37 \pm 2.20$ $\mu g/m^3$) was the highest WSI followed by $NH_4^+$ ($1.04 \pm 0.72$ $\mu g/m^3$) during the non-heating period, and the main reason is that Lanzhou is located on the transport path of the sandstorms from the Loess Plateau and Taklimakan Desert. Sandstorms occur frequently from March to May, carrying a large amount of dust with calcium carbonate ($CaCO_3$), as the main component [22,40,41]. Another reason is, that due to rising temperatures in spring, construction sites in Lanzhou gradually resume construction.

### 3.2. Overview of Air Quality and Influence Factors

According to the ambient air quality standards and the effects of various pollutants on human health, ecology and environment, the AQI simplifies the concentration of several air pollutants commonly detected into a single conceptual index, which is to express the trends of air quality. As shown in Figure 2, the AQI ranged from 36.17 to

149.13, with an average of 83.3. The average AQI was 104.09 during the heating period, which was about 1.5 times that of the non-heating period (67.71). Combined with China's secondary standards (GB 3095-2012), the pollution during the heating period was serious. The concentration of ions was strongly related to AQI ($R^2$ = 0.86). The highest AQI (160) occurred on December 11, 2016, and the ions concentration was the highest, as high as 89.99 μg/m³. The lowest AQI (36.17) appeared on August 20, 2017; the concentration of ions was relatively lower (11.35 μg/m³).

Meteorological conditions (e.g., WS and RH) have critical effects on the concentration of WSIs [42]. As shown in Figure 2, the WS ranged from 0.63 m/s to 1.84 m/s, with an average of 1.22 m/s. During the sampling period, the lower the WS, the higher the ion concentration, which suggested that high WS may be favorable for the diffusion of pollutants. For example, WS was increased from 1.6 m/s to 2.0 m/s on 11–27 December; the concentration of ions decreased from 89.99 to 37.54 μg/m³ conversely. On 6 April 2017, the concentration of WSIs reached 13.03 μg/m³, which was higher than the average (12.45 μg/m³) during the non-heating period, and it related to the maximum WS (1.84 m/s) and frequent sandstorms. RH plays a central role in the evolution and transformation of atmospheric aerosols, and the water content of aerosol mainly depends on RH and aerosol chemical composition. RH and the concentration of WSIs showed the same variation trends (Figure 2), which suggested that RH accelerated the chemical reaction and further increased secondary aerosols [43]. RH ranged from 19% to 95.29% (average of 47.8%). On December 11, 2016, RH and ions concentration both reached the highest values, which were 66.88% and 89.99 μg/m³, respectively. On 23 January 2017, the RH reached the lowest value (19%), the concentration of ions was at a low value (25.57 μg/m³). Above all, the influence of WS and RH on ion concentration should not be underestimated.

**Figure 2.** Temporal variation of water-soluble ions (WSIs) and meteorological factors.

*3.3. Secondary Transformation*

$SO_4^{2-}$ and $NO_3^-$ are usually the most crucial ions in atmospheric aerosols; $NO_x$ and $SO_2$ are precursor gases that form sulfates and nitrates. SOR and NOR are investigated to quantify the degree of conversion of $SO_2$ to $SO_4^{2-}$ and $NO_x$ to $NO_3^-$. Studies have shown that the critical values of SOR and NOR are both 0.1, and higher SOR and NOR values indicate higher secondary conversion efficiency [29]. As shown in Figure 3, SOR and NOR were 0.23 and 0.08 during the sampling period, SOR was higher than NOR, indicating the higher secondary transformation of $SO_2$ compared to $NO_2$. The $SO_4^{2-}$ concentration ($4.79 \pm 3.02$ μg/m$^3$) was lower than $NO_3^-$ ($6.60 \pm 8.06$ μg/m$^3$) and $SO_2$ ($23.79 \pm 17.5$ μg/m$^3$) was lower than $NO_2$ ($65.61 \pm 29.63$ μg/m$^3$). This might be due to the effective control of $SO_2$ emissions since the Chinese government implemented relevant measures, such as adjustment of industrial and energy structure, clean production, industrial desulfurization, green development et al. [44,45].

**Figure 3.** The variations of the sulfur oxidation ratio (SOR), nitrogen oxidation ratio (NOR), $SO_4^{2-}$, $NO_3^-$, $NH_4^+$, $SO_2$, $NO_2$ and $O_3$.

$SO_4^{2-}$ is mainly formed by homogeneous reaction. The main reason for the formation of sulfate from $SO_2$ is the influence of temperature and solar radiation, and the homogeneous oxidation of $SO_2$, OH radicals and $O_3$. Furthermore, the heterogeneous reaction is a function of RH [46]. During the heating and non-heating periods, the concentration of $SO_4^{2-}$ was 6.83 and 3.27 μg/m$^3$, and the concentration of $SO_2$ was 38.44 and 12.81 μg/m$^3$, respectively. During the non-heating period, SOR was higher (0.29) than in the heating period (0.16), which indicated that the high temperature contributes to the formation of $SO_4^{2-}$. The concentration of $O_3$ ranged from 29 to 197 μg/m$^3$ in Lanzhou during the non-heating period (average 125.2 μg/m$^3$) and was higher than those in the heating period. Combined with Figure 4a,b, we can find that higher $O_3$ concentration and temperature would favor $SO_2$ transformation to $SO_4^{2-}$ [47]. The formation of $SO_4^{2-}$ mainly experienced a homogeneous reaction in Lanzhou.

$NO_3^-$ is predominantly formed by homogeneous and heterogeneous reactions. When $NO_2$ reacts with gaseous $NH_3$ and is oxidized by OH to form $NH_4NO_3$, a homogeneous reaction occurs. Without OH, $NO_2$ is oxidized by $O_3$ to form $N_2O_5$ and then hydrates on the surface of particulate matter (PM) to form $NO_3^-$ [48]. During the heating and non-heating periods, the concentration of $NO_3^-$ was 12.20 μg/m$^3$ and 2.41 μg/m$^3$, respec-

tively. The concentration of $NO_2$ in the heating period (88.06 μg/m³) was higher than that in the non-heating period (48.77 μg/m³). NOR was 0.12 and 0.05, respectively, indicating the secondary transformation of $NO_2$ in the winter was more conspicuous. High temperature accelerated the decomposition of $NH_4NO_3$, resulting in weaker secondary transformation during the non-heating period [49]. As shown in Figure 4c,d, NOR exceed 0.1 and increased with RH under high RH conditions (RH = 35–60%), which indicated that $NO_3^-$ was mainly formed by heterogeneous reactions during the heating period, and high levels of $NO_3^-$ might be inferred as a function of $N_2O_5$ hydrolysis. Therefore, the lower temperature and high RH were more suitable for the formation of $NO_3^-$ and the heterogeneous reactions were dominant during the sampling periods.

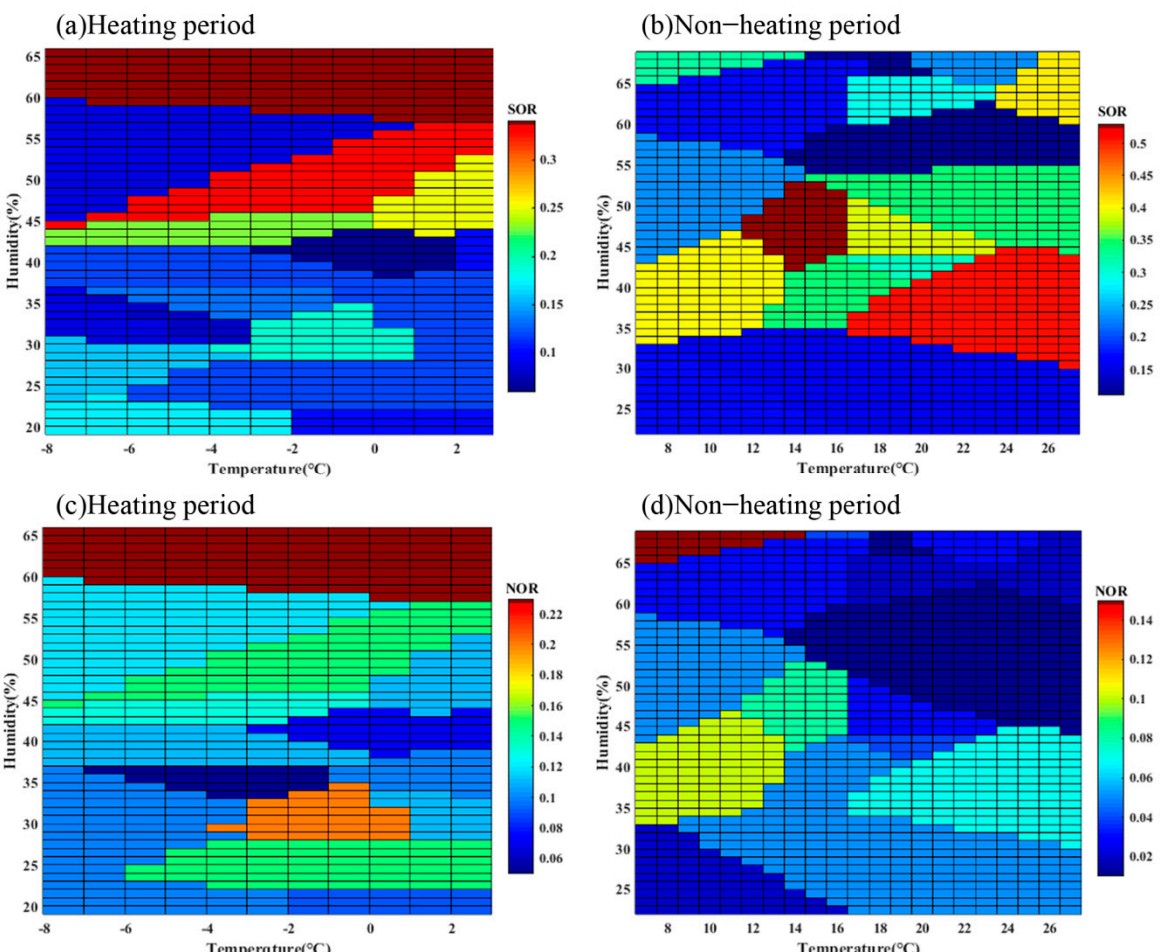

**Figure 4.** Relationship between SOR (**a**,**b**), NOR (**c**,**d**) and temperature and humidity (RH), respectively.

### 3.4. Long-Term Comparison of WSIs in Lanzhou

We compared our observed WSIs concentration with previous research on TSP in Lanzhou. By comparison, it could be concluded that $NO_3^-$, $SO_4^{2-}$, $NH_4^+$ and $Ca^{2+}$ were the predominant components, but the order of WSIs changed slightly in those years (Table 2). In 1990, $SO_4^{2-}$ and $Ca^{2+}$ were the two ions with the highest mass concentration. From 2005 to 2013, $SO_4^{2-}$ gradually increased. Since 2013, $SO_4^{2-}$ has decreased while the $NO_3^-$ has continued to increase, mainly due to the change of the industrial and energy structure and the increase in motor vehicles in Lanzhou. Recent studies have also found that the proportion of $NO_3^-$ presented a clear increasing trend as well as the increase in $NO_x$ emissions [50,51].

**Table 2.** Multi-year comparative analysis of WSIs in Lanzhou.

| Time | SO$_4^{2-}$ | NO$_3^-$ | Ca$^{2+}$ | NH$_4^+$ | Cl$^-$ | Na$^+$ | K$^+$ | Mg$^{2+}$ | Reference |
|------|------|-------|-------|-------|-------|------|------|------|-----------|
| 1990 | 57.10 | 9.69 | 11.79 | 10.72 | 3.16 | 4.03 | 2.04 | 1.47 | [52] |
| 2005 | 32.05 | 13.56 | 12.90 | 19.29 | 12.86 | 4.09 | 3.75 | 1.49 | [52] |
| 2007 | 32.35 | 13.51 | 15.06 | 18.08 | 11.26 | 4.21 | 3.93 | 1.60 | [52] |
| 2012 | 33.96 | 20.84 | 4.42 | 20.48 | 13.41 | 2.68 | 3.29 | 0.92 | [53] |
| 2013 | 38.76 | 18.81 | 9.50 | 19.38 | 6.40 | 2.45 | 3.48 | 1.22 | [53] |
| 2014 | 29.44 | 27.72 | 6.90 | 15.52 | 12.07 | 2.52 | 4.64 | 1.19 | [20] |
| 2016 | 20.88 | 29.27 | 24.93 | 13.57 | 5.43 | 2.76 | 1.84 | 1.34 | This study |

Figure 5 shows the energy consumption of industrial enterprises and the car ownership of urban residents in Lanzhou in recent years. The energy consumption mainly includes coal, coke, natural gas and crude oil. From 2012 to 2015, natural gas showed a significant decreasing trend. Coal and coke have shown a significant downward trend since 2014. In contrast, car ownership has been increasing, especially after 2013. It showed that the proportion of nitrogen oxides emitted by motor vehicle exhausts on air pollution has increased, and the proportion of sulfur dioxide produced by the combustion of fossil fuels on air pollution has decreased in recent years. With the change of energy structure and the rapid increase in the number of motor vehicles, atmospheric aerosol pollution in Lanzhou is changing to dust pollution and motor vehicle pollution.

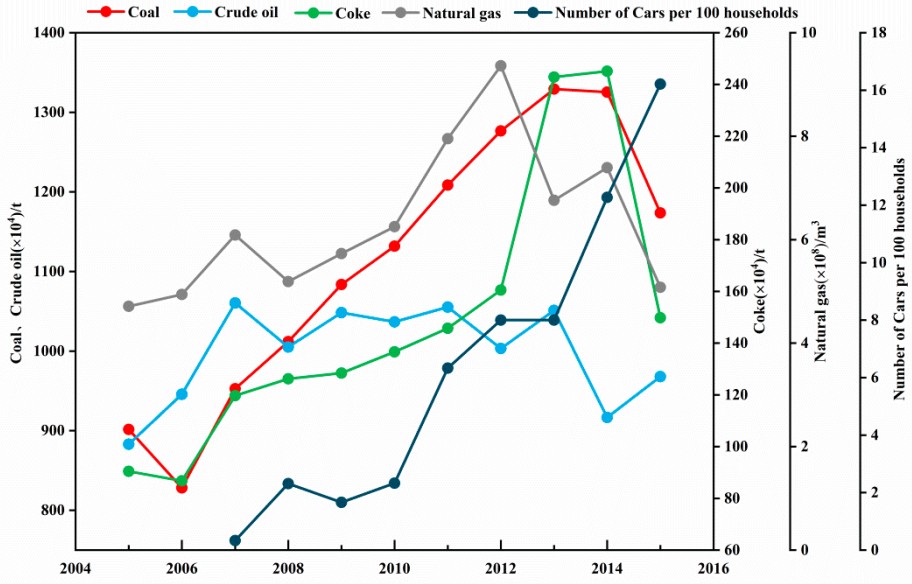

**Figure 5.** The energy consumption of industrial enterprises and the car ownership of urban residents in Lanzhou in recent years.

### 3.5. Local Sources

The clustering of components into a group indicates that the tested components may have come from a common source [54,55]. Based on hierarchical clustering analysis, the homology of eight WSIs was conducted with ORIGIN software. Ternary diagrams of [NO$_3^-$ + SO$_4^{2-}$ + NH$_4^+$], [Mg$^{2+}$ + Na$^+$ + Ca$^{2+}$] and [K$^+$ + Cl$^-$] are plotted in Figure 6. NO$_3^-$, SO$_4^{2-}$, NH$_4^+$ are identified as anthropogenic or second conversion source ions of WSIs [23]. Mg$^{2+}$, Na$^+$ and Ca$^{2+}$ are considered as tracers of crustal dust or natural sources, such as soil, dust and seawater droplets [56]. K$^+$ and Cl$^-$ are considered as biomass sources, Cl$^-$ is observed to be higher in winter, and is associated with elevated coal combustion emissions [57]. The proportion of anthropogenic source ions [NO$_3^-$ + SO$_4^{2-}$ + NH$_4^+$] was 0.61–0.75 during the heating period and was significantly higher than those in the non-heating period (0.23–0.45), indicating that the anthropogenic sources played an im-

portant role in the heating period. The proportion of natural source ions [$Mg^{2+}$ + $Na^+$ + $Ca^{2+}$] was 0.17–0.32 during the heating period and lower than in the non-heating period (0.50–0.75). The contribution of natural source ions was higher in the non-heating period due to the sandstorms occurring frequently. Lanzhou is an inland city in Northwest China, which is not vulnerable to the influence of salt particles. However, when the air mass passes through the adjacent Qinghai province, some salt particles in salt lakes may be brought into the atmosphere of Lanzhou. The proportion of biomass source ions [$K^+$ + $Cl^-$] was higher during the heating period (0.03–0.13) and lower in the non-heating period (0.01–0.1), indicating that the contribution of biomass source ions to WSIs was relatively stable and the increase in coal combustion obviously contributed to biomass source ions during the heating period. As shown in Figure 6, it was noted that a lower proportion of [$NO_3^-$ + $SO_4^{2-}$ + $NH_4^+$], a higher proportion of [$Mg^{2+}$ + $Na^+$ + $Ca^{2+}$] and [$K^+$ + $Cl^-$] were observed on 11 December 2016 and 31 January 2017. During the Chinese Spring Festival (31 January 2017), the increased concentrations of $K^+$ and $Cl^-$ might be due to the increased fireworks burning [58]. Meanwhile, the reduction in public travel during official holidays led to a reduction in traffic pollutant emissions; the proportion of [$NO_3^-$ + $SO_4^{2-}$ + $NH_4^+$] reached a lower value. On 11 December 2016, the concentrations of $Ca^{2+}$ (11.79 $\mu g/m^3$), $Mg^{2+}$ (0.68 $\mu g/m^3$) and $Na^+$ (1.42 $\mu g/m^3$) all exceeded the average values 6.82 $\mu g/m^3$, 0.39 $\mu g/m^3$, 0.84 $\mu g/m^3$, respectively, and reached higher levels in the heating periods. The contribution of [$Mg^{2+}$ + $Na^+$ + $Ca^{2+}$] to WSIs was significantly enhanced. According to the above analysis, the anthropogenic source ions and natural source ions were dominant ions of WSIs in the heating and non-heating periods, respectively. The biomass source ions were observed to be relatively stable.

**Figure 6.** Phase diagram of the ternary system: anthropogenic source ions: [$NO_3^-$ + $SO_4^{2-}$ + $NH_4^+$], natural source ions: [$Mg^{2+}$ + $Na^+$ + $Ca^{2+}$] and biomass source ions: [$K^+$ + $Cl^-$]. The green circle represents two special days (11 December 2016 and 31 January 2017) during the heating periods and the value tends to move to the low value area of the anthropogenic source ions and the high value area of the natural source ions.

### 3.6. Potential Source Aanalysis

To interpret the influence of regional sources and long-term transport of pollutants in Lanzhou throughout the sampling campaign, the 72-h air mass backward trajectories

ending were calculated by the HYSPLIT model and were classified into three clusters (T1–T3) both in the heating and non-heating periods (Figure 7). The concentration of pollutants of different air masses exhibited distinction due to the uneven distribution of emission sources around Lanzhou city. Thus, the mean concentration of WSIs for corresponding clusters of backward trajectories was summarized to investigate the pollutant levels from each air mass (Table 3).

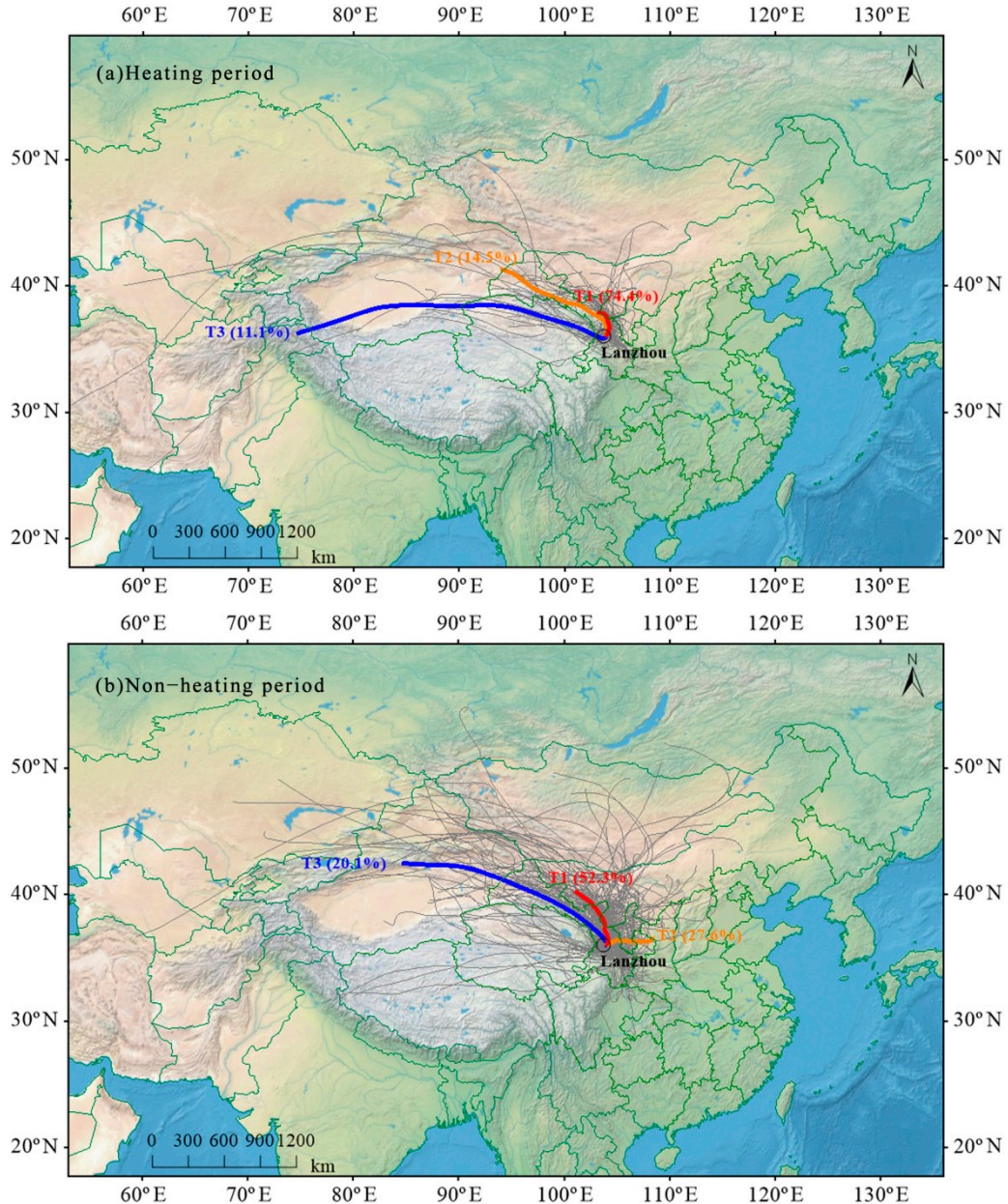

**Figure 7.** Five-day back backward trajectory cluster in Lanzhou.

A total of 74.4% of air masses were from the northeast of Lanzhou (T1) during the heating period, passing through the Wuwei and Baiyin cities. Baiyin city is the largest multi-variety nonferrous metal industrial base in China and releases sulfur dioxide and nitrogen oxides, and it has high loadings of $SO_4^{2-}$, $Ca^{2+}$. The dust from the Gobi Desert in Inner Mongolia and the pollutants from industrial emissions were carried to Lanzhou by the winter monsoon. The pervasive continental cold air masses also play an important

role in the T2 trajectory, accounting for 14.5% of the total air masses. This trajectory originated from the Xinjiang province and throughout the Hexi Corridor. Combined with Table 3, the concentration of $NH_4^+$ was the highest of the total trajectories, because of the biomass combustion (animal husbandry) in Gansu province. T3 trajectory was mainly derived from Central Asia (accounting for 11.1%), passing through Taklimakan, Qaidam Basin and Qinghai salt lake. This trajectory was characterized by long-distance transmission, the concentration of $NO_3^-$ obviously increased.

During the non-heating period, the air masses that arrived in Lanzhou were different from the heating period. It can be seen that the northern transport pattern (T1) was frequent and accounted for 52.3% of total air masses from Figure7. T1 generally originated from Tengger Desert, which is situated on the Inner Mongolia Plateau, and traveled into Wuwei and Baiyin city. The highest contributions of $NO_3^-$, $NH_4^+$, $Ca^{2+}$ and $Cl^-$ to the WSIs in all the air masses were acquired in T1, suggesting that T1 was dominant in the contribution of pollutants to Lanzhou in the non-heating period. Under the influence of the southeast monsoon, the T2 trajectory originated from Shaanxi province, which only accounted for 27.6% of total clusters in the non-heating period, followed by those from the Xinjiang province passing through the Turpan Basin and Hexi Corridor (T3, 20.1%), which represented the long-range air masses. It should be noted that all the clusters carried a higher value of $Ca^{2+}$, which may be related to the specific natural environment. The dust from the Gobi Desert around Lanzhou contributed to the $Ca^{2+}$ of WSIs.

**Table 3.** Statistics of the average concentration of WSIs for each type of air mass arriving in Lanzhou.

| Period | Air Mass Type | $SO_4^{2-}$ | $NO_3^-$ | $NH_4^+$ | $Ca^{2+}$ | $Cl^-$ | $Na^+$ | $Mg^{2+}$ | $K^+$ |
|---|---|---|---|---|---|---|---|---|---|
| | T1 | 7.93 | 4.36 | 3.84 | 7.00 | 0.97 | 0.98 | 0.36 | 0.37 |
| Heating period | T2 | 5.47 | 8.60 | 8.73 | 6.90 | 2.57 | 0.79 | 0.40 | 0.70 |
| | T3 | 6.85 | 12.91 | 5.79 | 6.80 | 2.16 | 0.83 | 0.39 | 0.66 |
| | T1 | 3.23 | 2.99 | 1.16 | 4.96 | 0.60 | 0.74 | 0.25 | 0.25 |
| Non-heating period | T2 | 2.87 | 2.21 | 0.94 | 3.64 | 0.43 | 0.22 | 0.16 | 0.22 |
| | T3 | 3.54 | 2.21 | 1.04 | 4.51 | 0.51 | 0.36 | 0.22 | 0.25 |

Combined with the concentration of six standard pollutants ($SO_2$, $NO_2$, $CO$, $O_3$, $PM_{2.5}$ and $PM_{10}$) in Lanzhou City, the potential source areas and contributions of the pollutants were analyzed by using PSCF and CWT analysis methods. It is worth noting that the concentration of $PM_{10}$, $PM_{2.5}$ and $NO_2$ exceed the China's secondary standards (GB 3095-2012), and account for 56%, 56%, and 67% of the total sampling period of the heating periods, respectively. Here, we mainly chose $PM_{10}$, $PM_{2.5}$ and $NO_2$ as the main pollutants during the heating period in Lanzhou. Figure 8 shows the PSCF and CWT analysis of $PM_{10}$ (a, b), $PM_{2.5}$ (c, d) and $NO_2$ (e, f), respectively. For $PM_{10}$, the PSCF primarily identified sources from the surrounding area, as northwest wind during the campaign period was relatively frequent. The highest PSCF values for $PM_{10}$ were contributed in the Hexi Corridor, northeastern Qinghai–Tibetan Plateau, northern Qinghai province and Inner Mongolia Plateau; all of the probability was greater than 80%. Meanwhile, it can be seen from CWT that the contribution of these regions to $PM_{10}$ in Lanzhou was between 120 and 270, indicating many air pollutants transported from these areas with the Gobi Desert. $PM_{2.5}$ and $NO_2$ showed similar potential source regions and were mainly distributed by pollution from the Hexi Corridor and northeastern Qinghai–Tibetan Plateau. In addition, the cities surrounding Lanzhou such as Wuwei and Baiyin made greater contributions to $PM_{10}$, $PM_{2.5}$ and $NO_2$. Lanzhou is a typical industrial city with serious pollutant emissions, the $PM_{10}$, $PM_{2.5}$ and $NO_2$ concentrations were mainly influenced by local fossil fuel combustion.

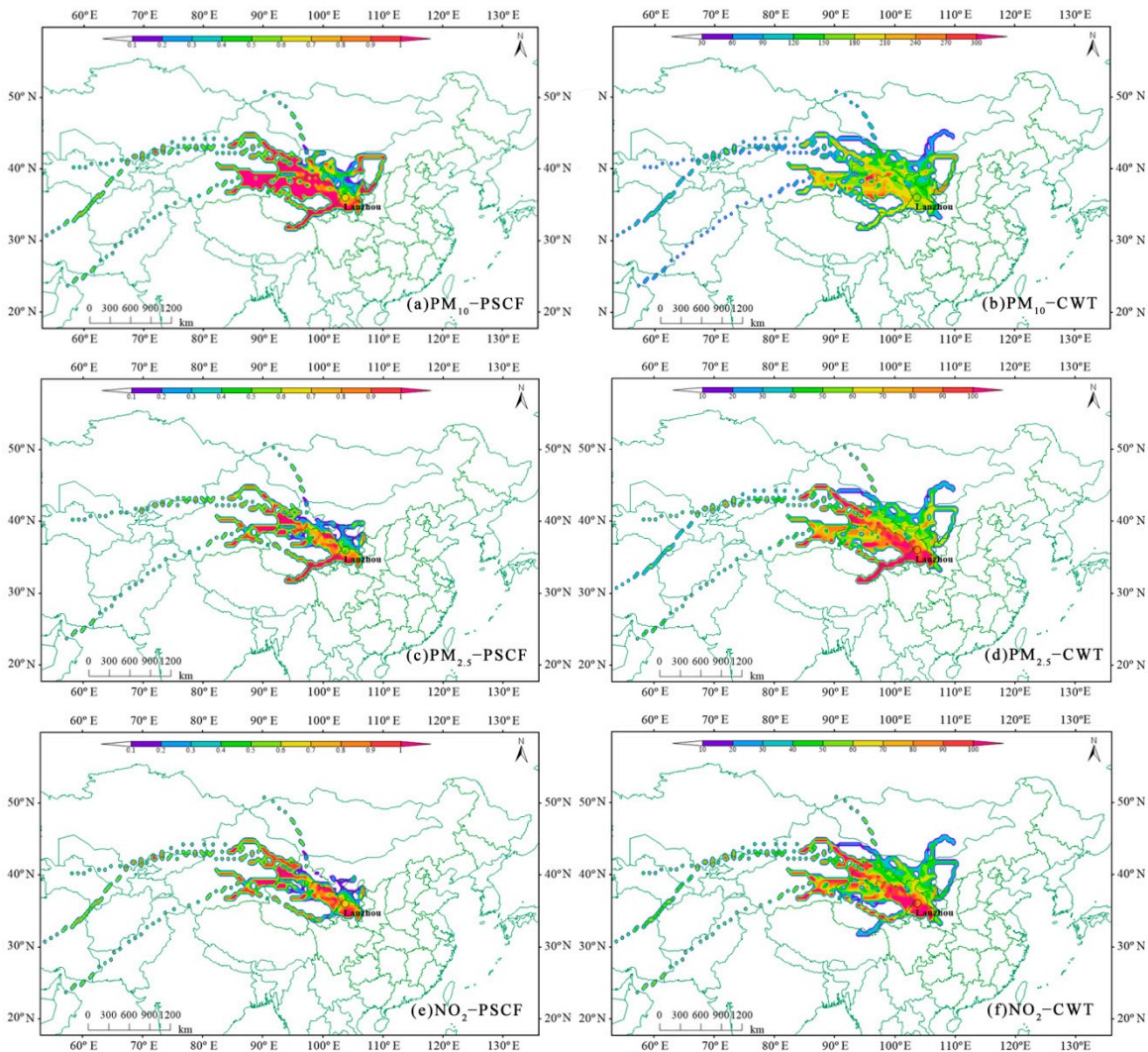

**Figure 8.** Potential source contribution function (PSCF) of particulate matter (PM)$_{10}$ (**a**), PM$_{2.5}$ (**c**) and NO$_2$ (**e**) and concentration-weighted trajectory (CWT) of PM$_{10}$ (**b**), PM$_{2.5}$ (**d**) and NO$_2$ (**f**) in the heating period in Lanzhou.

## 4. Conclusions

The aerosol samples were collected from 2016 to 2017 in Lanzhou, a semi-arid and chemical-industrialized city, Northwest China. We investigated the characteristics, secondary transformation and the potential sources of water-soluble ions (WSIs: NO$_3^-$, SO$_4^{2-}$, Na$^+$, NH$_4^+$, K$^+$, Mg$^{2+}$, Ca$^{2+}$, Cl$^-$). The concentration of WSIs was higher in the heating period (35.68 ± 19.17 μg/m$^3$) and lower in the non-heating period (12.45 ± 4.21 μg/m$^3$). NO$_3^-$, SO$_4^{2-}$, NH$_4^+$ and Ca$^{2+}$ were dominant WSIs. WSIs concentration had a strong correlation with the AQI and meteorological factors; for example, low WS and high RH contributed to the formation of the WSIs.

Compared to previous studies, the concentration of SO$_4^{2-}$ had decreased, while the NO$_3^-$ level was increasing, probably due to the change of energy structure and the proportion of vehicular exhaust increased. The average SOR and NOR values exceed 0.1, which verified that SO$_4^{2-}$ and NO$_3^-$ were suffering more secondary transformation. The formation of SO$_4^{2-}$ mainly experienced a homogeneous reaction with higher O$_3$ concentration and temperature. The lower temperature and high RH were more suitable for the formation of NO$_3^-$ and the heterogeneous reactions were dominant during the sampling periods.

Three-phase cluster analysis showed that the anthropogenic sources [$NO_3^-$ + $SO_4^{2-}$ + $NH_4^+$] and natural sources [$Mg^{2+}$ + $Na^+$ + $Ca^{2+}$] were the dominant ions of WSIs in the heating and non-heating periods, respectively. The results of the backward trajectory analysis and the potential source contribution function model indicated the Lanzhou was strongly influenced by the Hexi Corridor, northeastern Qinghai–Tibetan Plateau, northern Qinghai province, Inner Mongolia Plateau and its surrounding cities.

**Author Contributions:** Data collection, X.Z. (Xi Zhou) and S.M.; formal analysis, H.J.; funding acquisition, Z.L.; investigation, F.W. (Fanglong Wang); software, H.J.; writing—original draft, H.J.; methodology, X.Z. (Xin Zhang); writing—science advising, review, F.W. (Feiteng Wang). All authors have read and agreed to the published version of the manuscript.

**Funding:** This work is supported by the Second Tibetan Plateau Scientific Expedition and Research (2019QZKK0201), the Strategic Priority Research Program of Chinese Academy of Sciences (Class A) (XDA20060201; XDA20020102), the National Natural Science Foundation of China (41761134093; 41471058) and the SKLCS founding (SKLCS-ZZ-2020). We gratefully thank the reviewers for their constructive comments.

**Institutional Review Board Statement:** Not applicable.

**Informed Consent Statement:** Not applicable.

**Data Availability Statement:** Not applicable.

**Acknowledgments:** We thank the staff working in the Tianshan Glaciological Station for helping collect data. We also gratefully thank the reviewers for their constructive comments.

**Conflicts of Interest:** The authors declare no conflict of interest.

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
