# Peer review of "Water-Soluble Ions in Atmospheric Aerosol Measured in a Semi-Arid and Chemical-Industrialized City, Northwest China"

_atmosphere, doi:10.3390/atmos12040456_

Round 1
Reviewer 1 Report
The manuscript represents water-soluble ions measurements in Northwest China. The manuscript represents measurements mainly from the year 2017.
I have a few minor remarks.
What instrument was used for the aerosol particle collection? What was flow rate and other parameters? What kind of separator was used (PM10, PM2.5?) Does different size particles were investigated, or only total suspended particles were collected? In the whole text there is TSP, only in end of the section 3.5 PM10, PM2.5 are appearing. Is not clear, concentrations of PM10 and PM2.5 were measured by the Authors, or taken from local air quality stations?
Section 3.3. "... since the implementation of relevant measures". What king of measures were implemented?
Section 3.4. "... account of the drastic measured carried out by the Chinese government." The citation must be provided showing the measured carried out.
Caption of the Fig 6 must be extended. The meaning of the dotted green circle must be described.
Author Response
We thank the reviewer for the comments and questions. We wrote the reply in the attached file (Responses to Reviewer 1.pdf).
Thanks again,
Best regards.

Reviewer 2 Report
This is a good study and can be accepted after some minor revisions.
- There should be a space before and after the sign “±”. Correct it throughout the ms.
- It should be “µg m-3”. Correct it throughout the ms.
- Abstract: “was effect” changes to “was affected”
- “ion chromatographer” is not a technical word. Change it to “an ion chromatograph”.
- In Table 1: Generally, ∑+ and ∑- are used for cation and anion charge equivalent, respectively. So do not use these to report mass concentration. It is confusing. Simply write total ions.
Good luck.
Author Response
We thank the reviewer for the comments and questions. We wrote the reply in the attached file (Responses to Reviewer 2.pdf).
Thanks again,
Best regards.
